# Formation of Subwavelength Periodic Triangular Arrays on Tungsten through Double-Pulsed Femtosecond Laser Irradiation

**DOI:** 10.3390/ma11122380

**Published:** 2018-11-26

**Authors:** Hongzhen Qiao, Jianjun Yang, Jing Li, Qi Liu, Jie Liu, Chunlei Guo

**Affiliations:** 1School of Electronic and Electrical Engineering, Shangqiu Normal University, Shangqiu 476000, China; qiaohongzhen8@163.com (H.Q.); sunnyweather@163.com (J.L.); 2State Key Laboratory of Applied Optics, Changchun Institute of Modern Optics, Fine Mechanics and Physics, Chinese Academy of Sciences, Changchun 130033, China; 3Key Laboratory of Photochemical Conversion and Optoelectronic Materials, Technical Institute of Physics and Chemistry, Chinese Academy of Sciences, Beijing 100190, China; lijingsp@mail.ipc.ac.cn; 4Institute of Modern Optics, Nankai University, Tianjin 300071, China; TJliuqi_67@163.com; 5The Institute of Optics, University of Rochester, Rochester 14627, NY, USA

**Keywords:** time-delayed femtosecond lasers, orthogonal polarizations, subwavelength triangular structures, tungsten

## Abstract

We present a mask-free strategy for fabricating two-dimensional subwavelength periodic triangular arrays on tungsten, by focusing two orthogonally polarized and temporally delayed femtosecond laser beams using a cylindrical lens. In stark contrast to the commonly observed structures of either a single ablation spot or a one-dimensional grating, we obtained highly uniform periodic triangular arrays on the laser-exposed surface, with three equilateral sides each of 480 nm in length and about 100 nm in modulation depth. The triangular features varied with both the laser energy and the scanning speed. We found that the optical reflectivity of such a surface reduces significantly within the spectral range of 700–2500 nm. The triangular structure morphology can also be controlled by varying the time delay between the two laser beams.

## 1. Introduction

The study of micro- and nanoscale structuring of metal surfaces has grown rapidly in recent years because of the effects in altering the surface properties, which promises unconventional applications in areas such as plasmonics [1], sensing [2], catalysis [3], broadband light absorption [4,5], and wettability [5,6]. Usually, the fabrication of material microstructures is based on several mature technologies including charged-beam lithography [7,8,9], nano-imprinting [10], and laser direct/interference writing [11,12,13,14], which often require costly and sophisticated equipment with complex multiple procedures as well as being time consuming. Recently, femtosecond lasers have emerged as a novel tool for producing micron and nanometer structures on a large variety of materials, especially where the possibility of surface processing on metal targets is based on the pulse ablation (i.e., material ejection) from the irradiated sample. As a universal phenomenon of laser-matter interaction, femtosecond laser-induced periodic surface structures (LIPSS), has been identified as having the potential of large-area structuring material surfaces within one step by the formation of a group of one dimensional (1D) (quasi-) parallel gratings at subwavelength or even deep subwavelength scales [15,16,17], in sharp contrast to the tiny interaction spot induced by two-photon lithography with the essential help of tight focusing conditions [18,19].

Because of potentials in both fundamental research and applications, numerous efforts have strived for the generation of 1D LIPSS with variable laser parameters, material properties, and surrounding environments, which try to understand the underlying mechanisms [20,21,22,23,24]. Nevertheless, until now the quality of such laser-induced surface structures still needs to improve. On the other hand, by adopting two temporally delayed femtosecond laser beams with orthogonal polarizations, we reported the direct fabrication of two-dimensional (2D) periodic subwavelength structures on the metal surfaces [25,26], where the formation of the structure units with either circular or elliptical outlines are attributed to the independent excitation of surface plasmons from the two laser beams. In fact, such phenomena can also be considered as a linear superposition of two crossed 1D LIPSSs. Because the subwavelength structures with complex geometry are attractive in controlling electromagnetic waves [27,28], advancing the technique of femtosecond LIPSS to flexibly produce novel structures especially on metal targets becomes a much needed technique.

In this paper, we present a one-step-process method by using two orthogonally polarized femtosecond laser beams (800 nm, 50 fs, 1 kHz), to directly obtain 2D periodic arrays of subwavelength triangular structures on the hard surfaces of tungsten without any masks. The structure units display triangular profiles with side-lengths of about 480 nm, and their formation was investigated in terms of the scanning speed and the laser energy. The effects of the triangular structures on the optical reflectivity of the metal surface were also measured. Finally, we found that the time delay of the two femtosecond laser beams can be used to control the triangular structures.

## 2. Experiment

First, we demonstrate the formation of the periodic arrays of subwavelength triangular structures on tungsten surfaces, by using a compact experimental setup to generate two orthogonally polarized femtosecond laser beams at a short time delay, as shown in Figure 1a. A chirp-pulsed amplification of Ti: sapphire femtosecond laser system (Spitfire Ace, Spectra physics, Santa Clara, CA, USA) was employed as an irradiation source, which delivers the horizontally polarized pulse trains at 1 kHz repetition rate, with a central wavelength and laser pulse width of 800 nm and 50 fs, respectively. A 1.86 mm thickness of YVO_4_ birefringent crystal was employed to temporally split each femtosecond laser pulse into two, the collinear propagation of which are linearly polarized in orthogonal directions and have Δt = 1.2 ps separation in the temporal domain. The azimuth angle between the incident laser polarization and the optical axis of the crystal was set as θ = 42°. Then the two femtosecond laser beams were focused by one plano-convex cylindrical lens (focal length of f = 50 mm), leading to an elliptical-shaped focal beam spot. A bulk tungsten plate with dimensions of 25 × 25 × 1 mm^3^ served as a material sample to be micro-structured, whose surfaces were mechanically polished with a fine grade of emery paper to a surface roughness of about Ra = 4.09 nm and degreased in acetone before the experiments. The metal sample was mounted on a computer-controlled three-axis translation stage (ESP301, Newport Inc, Irvine, CA, USA) with a resolution of 1 μm. In order to avoid strong laser damage, the target surface was moved 0.2 mm away from the focus towards the lens. The incident laser energy onto the sample was controlled by neutral-density attenuators. The surface microstructures were obtained by translating the sample perpendicular to both the laser beam propagation and the major axis of the elliptical beam spot in an ambient air environment. The laser-exposed surface morphologies were examined by scanning electron microscopy (SEM, Phenom, Eindhoven, The Netherlands) and atomic force microscopy (AFM, Bruker, Billerica, MA, USA).

## 3. Results and Discussions

Figure 2a shows the typical SEM images of the surface morphology induced by the orthogonal polarization of two femtosecond laser beams at a total energy of E = 0.18 mJ, with the sample scanning speed given by 0.03 mm/s. In contrast to the commonly observed 1D grating structures in the previous reports [17,20,23,24], a new type of 2D periodic structures appears on the laser-exposed surface, where the subwavelength-scaled structure units displaying the evident triangular profiles are well organized on the metal surface. The detailed inspections reveal that each triangular unit is in fact constructed by spatial interlocking of three ablation grooves, or the 2D periodic triangular structure arrays are developed by crossing of three groups of grating-like LIPSSs oriented in different directions, where each group exhibits periodicity of about 610 nm. Extremely perplexing, the number of the grating LIPSS is more than that of the incident femtosecond laser beams, which indicates a nonlinear process occurring during the structure formation. The measured three side lengths of the structure unit are equivalent to about 480 nm, much smaller than the incident laser wavelength. In other words, the intersection angles between any of two grating structures are 60°. From the high resolution SEM picture, we can find the formation of a regular hexagonal pattern by six adjacent triangular units, whose center is located at the crossing point of three ablation grooves. The width of each groove measured only 130 nm. Noticeably, such exquisite surface structures are in fact self-organized within one step rather than by the point-to-point laser ablation processes. The corresponding AFM measurement results, as shown in Figure 2b, reveal that the modulation depth of the subwavelength triangular structures is approximately 100 nm.

In fact, the formation of such periodic triangular structure arrays was also investigated under other experimental conditions. Figure 3a,b shows the obtained results for two different scanning speeds of 0.02 mm/s and 0.03 mm/s at a total incident laser energy of E = 0.18 mJ. For the higher scanning speed, the structure dimensions seem to be larger with more distinct and uniform appearance or we can understand that in this case the laser-induced ablation grooves become more regular and spatially extended, being different from the random and fragmental distribution in the previous reports especially with the single-beam laser irradiation [17,20]. If the scanning speed continuously increased, however, the formation of the subwavelength triangular surface structures began to deteriorate.

Figure 3c,d shows the obtained results for two different incident laser energies of E = 0.1 mJ and 0.16 mJ at a scanning speed of 0.03 mm/s. In the case of small laser energy, the triangular outlines of the structures begin to be degraded, or the distinctness of the three groups of the ablation grooves becomes unequal, which results in a predominance of the ablation grooves with orientation from the upper right to the lower left. On the other hand, in the case of large laser energy, the subwavelength triangular structure formation presents clear and regular distribution, and the sharpness of the ablation grooves with different orientations appears to be identical.

In order to characterize the influence of the subwavelength triangular structures on the optical properties of the tungsten surface, the spectral reflectivity of the structured surface was measured using a Fourier transform infrared spectrometer (VERTEX-70, Bruker, Leipzig, Germany), and the obtained results are shown in Figure 4, where the optical reflectivity of the polished sample is also given for comparison. It can be seen that after the femtosecond laser processing the optical reflection of the tungsten surface is greatly decreased within a wavelength range of 700–2500 nm. The shorter the wavelength, the more decreased the reflection becomes. In particular, a reflection dip begins to appear near the wavelength of 1.0 µm, about five times smaller than that of the polished surface, which suggests the spectral modulation effect of the subwavelength triangular structures. According to ref. [17,29], the measured decreasing optical reflection also indicates enhancements in the optical absorptivity and thermal emission of the material surface.

During the aforementioned experiments, although a birefringent crystal was adopted for the generation of two femtosecond laser beams in a simple and compact way, the time delay between the two laser beams could not be continuously altered. To gain deep insight into the formation processes of such surface structures, an additional experiment was performed with a Mach–Zehnder interferometer to generate two femtosecond laser beams with variable time delays, as shown in Figure 1b. Under such circumstance two orthogonally polarized (respectively along the vertical and horizontal directions) laser pulses were achieved by inserting a half wave-plate in one of the optical paths. After certain temporal delays, the two femtosecond laser beams were collinearly overlapped for the beam focusing through one cylindrical lens. The energies of the two laser beams linearly polarized in the vertical and horizontal directions were set as E_h_ = 0.082 mJ and E_v_ = 0.068 mJ, respectively. As shown by Figure 5, when the time delay is Δt = −5 ps, i.e., the striking time of the vertically polarized laser pulse onto the target is earlier than that of the horizontally polarized one, the uniform 2D subwavelength triangular structures can be generated on the metal surface. Among the three groups of ablation grooves, one is oriented in the vertical direction, being perpendicular (or parallel) to the polarization direction of the delayed (or prior) incident laser pulse, and other two are slantwise oriented regarding any laser polarizations.

As the time delay increased to Δt = −40 ps, the triangular structures were still found on the sample surface. In comparison with the observation in Δt = −5 ps, the spatial orientations of the ablation grooves can still be maintained but the structure sharpness is reduced. For example, some nanoscale villi form substances occur inside the ablation grooves; moreover, in some places the slantwise oriented grooves appear to be discontinuous. Remarkably, this phenomenon becomes pronounced at the larger time delay of Δt = −80 ps when the physical correlations between the two laser striking processes are weakened. In this case the laser-induced surface structures are degraded into a predomination of 1D grating patterns with orientation in the vertical direction. According to previous studies [17,20], orientation of the grating structures induced by femtosecond laser on metal surfaces are usually perpendicular to the direction of the laser polarization. Therefore, in our cases the development of the vertically oriented grooves can be attributed to the temporally delayed incident laser of the horizontal polarization, and the formation of the other two slantwise oriented grooves can also be considered to be from modified interactions of the delayed laser pulse (horizontally polarized) by the transient properties of the metal surface optically excited by the prior laser pulse (vertically polarized). In other words, the incident temporally delayed incident laser pulse seems to be responsible for the emergence of the three ablation grooves with different orientations to constitute the triangular-profiled surface structures.

The validity of this assumption was confirmed by a further experiment with a positive time delay between two laser beams, where the irradiation of the vertically polarized laser pulse was temporally delayed; Δt = 5 ps regarding the horizontally polarized one. The obtained results are shown in Figure 5d. Clearly, the periodic triangular structure arrays can still appear on the metal surface, however, the spatial orientations for the three groups of ablation grooves become very different from those at the negative time delays, i.e., the periodic ablation of grooves oriented in the vertical direction disappears, instead the parallel grooves with the horizontal orientation began to form, which is usually attributed to interference between the delayed incident vertical polarization of the laser pulse and its excited surface plasmon [17,20,26]. The two groups of the slantwise oriented grooves may be due to nonlinear modulation effects of the transient surface dynamics triggered by the prior incident laser pulse (horizontally polarized). A brief explanation of the nanotriangular structure formation is described as follows. In the case of the two time-delayed femtosecond laser pulses, the irradiation of the first laser pulse can lead to the spatially periodic change of the surface properties by the distribution of the laser intensity fringes resulting from interference of the incident light with the excited surface plasmons. This can transiently alter optical properties such as refractive index of the surface into a spatially periodic pattern, named as a transient index grating. When the second laser pulse is incident to the material surface, especially with the direction of the laser polarization perpendicular to a grating vector of the transient index change, the excitation of the surface plasmons can be modified because of the non-equilibrium properties and destroyed translational symmetry of the metal surface [30]. As a consequence, apart from the surface plamson excitation along the direction of the second laser polarization, two other excitations can be also generated along different directions when considering the nonlinear coupling between two collective motion behaviors of the surface electrons, triggered by two laser pulses with different linear polarizations. Correspondingly, three groups of parallel nanogrooves are finally observed on the material surface. Of course, clear understanding of the physical mechanisms for the aforementioned phenomena will be explored in the future.

## 4. Conclusions

In summary, we introduced a mask less method for the fabrication of 2D periodic triangular structure arrays on the hard material surface of tungsten, by irradiation of two time-delayed femtosecond laser beams with orthogonal polarizations through different experimental configurations. First, the generation of two femtosecond laser beams at a fixed time delay of 1.2 ps was obtained with a birefringent crystal, and the laser-exposed surface displayed three groups of parallel nanogrooves oriented in different directions to constitute the periodic triangular structure arrays. The triangular units were measured to have side-lengths of about 480 nm and a modulation depth of near 100 nm. Through varying the laser energy and the scanning speed, we obtained the optimal conditions for the formation of such triangular structures. The spectral reflectivity of the structured metal surface was found to be greatly reduced in the wavelength range of 700–2500 nm, with minimum value about five times smaller than that of the untreated surface.

Second, the two femtosecond laser beams with orthogonal polarizations and tunable time delays were generated by using an interferometer-like setup. For time delays less than 40 ps, triangular structures can be uniformly produced on the metal surface, but were readily degraded into 1D ripple structures at larger time delays. In addition, the spatial alignment of the triangular structures was confirmed to depend closely on the direction of the linear polarization of the delayed incident laser beam in the experiment. Our experimental investigations promise a rapid and flexible way to develop an optical metasurface on metal materials for potential applications in nanophotonics.

## Figures and Tables

**Figure 1 materials-11-02380-f001:**
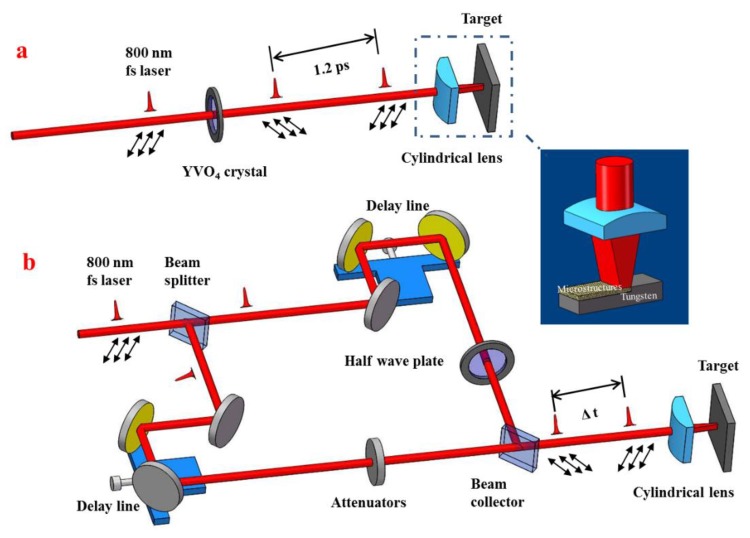
This two schematic diagrams for the formation of 2D subwavelength periodic triangular structures on tungsten surfaces using two time-delayed femtosecond laser beams with orthogonal polarizations. (**a**) generation of two femtosecond laser beams with a fixed time delay of 1.2 ps via a birefringent crystal; (**b**) generation of two femtosecond laser beams with tunable time delays based on an interferometric configuration. The double arrows represent directions of the linear polarization of femtosecond laser beams.

**Figure 2 materials-11-02380-f002:**
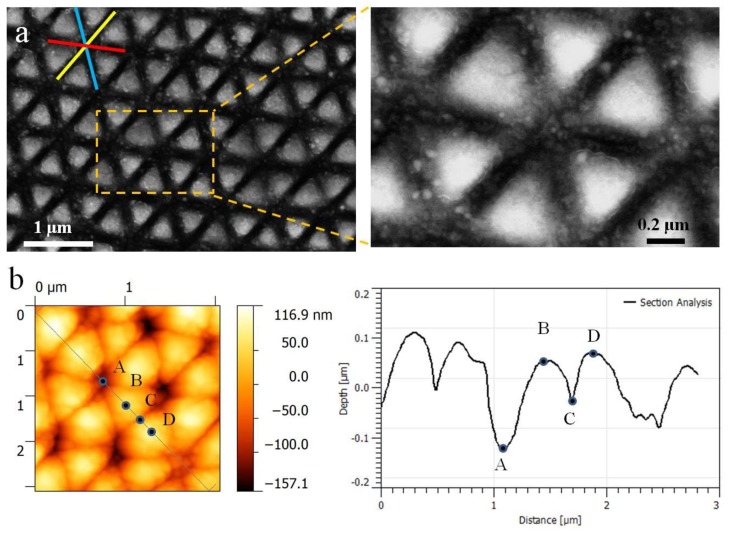
(**a**) SEM images of 2D periodic subwavelength triangular structures on tungsten surfaces induced by two time-delayed femtosecond laser beams with orthogonal polarizations. The laser energy and the sample scanning speed are E = 0.18 mJ and 0.03 mm/s, respectively; (**b**) AFM measurement results (including a cross-section profile) of the laser-induced triangular surface structures, where A, B, C, and D represent different positions on the structure surface.

**Figure 3 materials-11-02380-f003:**
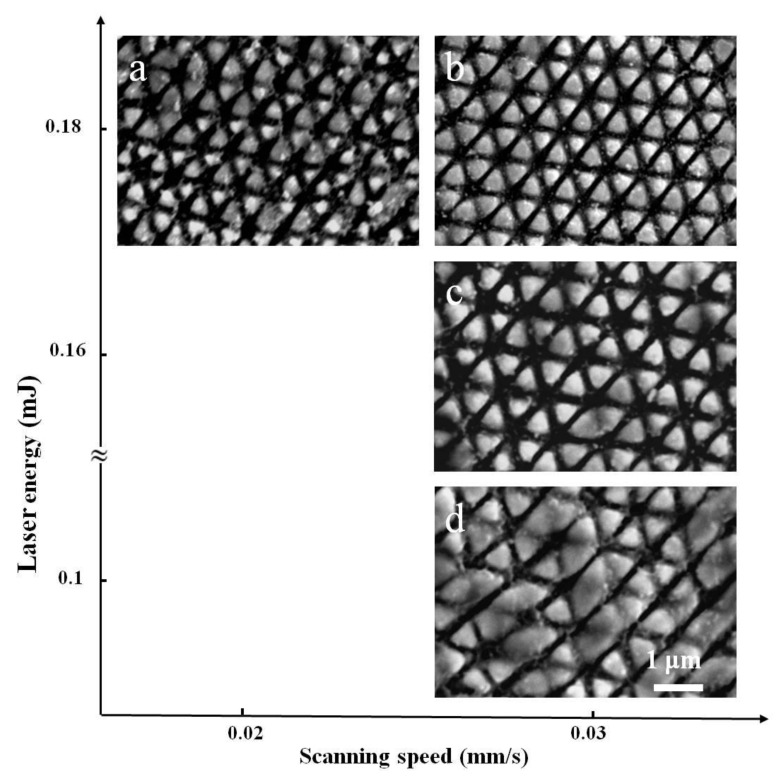
(**a**) represents the results under the sanning speed of v = 0.02 mm/s and the energy of E = 0.18 mJ; (**b**) for the conditions of v = 0.03 mm/s and E = 0.18 mJ; (**c**) for the conditions of v = 0.03 mm/s and E = 0.16 mJ; (**d**) for the conditions of v = 0.03 mm/s and E = 0.1 mJ.

**Figure 4 materials-11-02380-f004:**
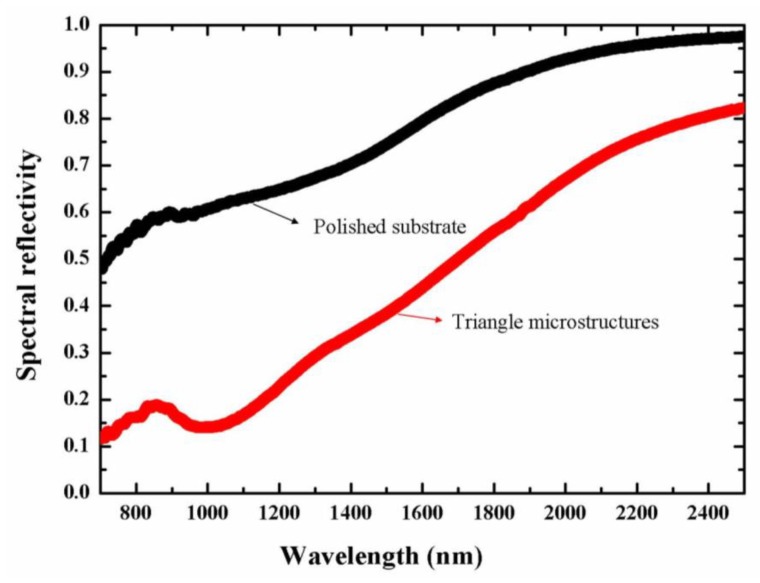
Measured reduction of the optical reflectivity for tungsten surfaces covered with subwavelength triangular structures (red solid line) in comparison with the polished surface (black solid line).

**Figure 5 materials-11-02380-f005:**
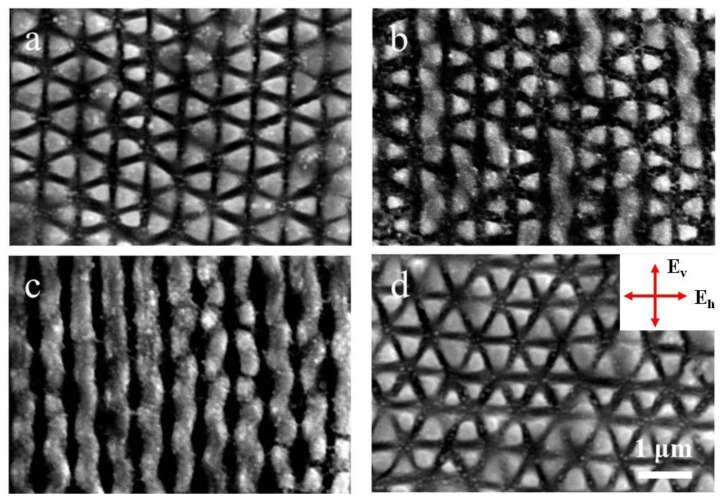
Morphological changes of the tungsten surface irradiated by double-pulsed femtosecond lasers at variable time delays. (**a**) Δt = −5 ps, (**b**) Δt = −40 ps, (**c**) Δt = −80 ps, and (**d**) Δt = 5 ps. The energies of the two femtosecond lasers are E_h_ = 0.082 mJ and E_v_ = 0.068 mJ, respectively. Their linear polarization directions are denoted by the red double arrows.

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
