# Peer review of "Formation of Subwavelength Periodic Triangular Arrays on Tungsten through Double-Pulsed Femtosecond Laser Irradiation"

_materials, 2018, doi:10.3390/ma11122380_

Reviewer 1 Report

     Fabrication of a 2D triangular pattern with a femtosecond laser is presented in the manuscript. The results are very interesting and publishable. However, the authors did not explain their results at all. Thus, I recommend that the authors revise their manuscript.

      My specific comments are below:

     1. What is the specification of the beam collector? What is the grade of the polishing paper? Grit No.? What is the surface roughness?

     2. This observation of triangular structure is very important. However, could the authors report the threshold fluence for disappearance and appearance of this type of structure?

     3. Why did not the authors measure the wettability regarding contact angle? Contact angle measurement will be very useful for the application of such textures.

     4. This manuscript is sorely lacking the explanation on the formation of triangular structure. The authors did not even try to explain this phenomenon based on existing literature. Why giving a single line explanation with a promise to explore it in future? Please elaborate the explanation based on the laser-material interaction literature. See some review articles for reference: doi:10.3390/mi5041219, doi:10.1002/lpor.201200017.

Minor comment: Line 215: not very unclear or not very clear?

Reviewer 2 Report

Please find my report attached.

Reviewer 3 Report

Dear Authors,

       Congratulations for these fine results. From my point of view, this paper can be bublished as it is. Even if the final explanation of the processes involved in forming the observed surface structures is left open for further investigation, the paper is a very goog report on observations that might give a base for further investigations of this topic.  

Author Response

No further comments for our responses.

Round  2

Reviewer 1 Report

       The authors answered all the questions raised in the first review. I recommend the manuscript for publication.